# RNAmigos2: accelerated structure-based RNA virtual screening with deep graph learning

Juan G. Carvajal-Patiño [1,2,10], Vincent Mallet[3,4,5,6,10], David Becerra[1,2], Luis Fernando Niño Vasquez[2], Carlos Oliver[7,8,9,11] & Jérôme Waldispühl [1,11]

RNAs are a vast reservoir of untapped drug targets. Structure-based virtual screening (VS) identifies candidate molecules by leveraging binding site information, traditionally using molecular docking simulations. However, docking struggles to scale with large compound libraries and RNA targets. Machine learning offers a solution but remains underdeveloped for RNA due to limited data and practical evaluations. We introduce a data-driven VS pipeline tailored for RNA, utilizing coarse-grained 3D modeling, synthetic data augmentation, and RNA-specific self-supervision. Our model achieves a 10,000x speedup over docking while ranking active compounds in the top 2.8% on structurally distinct test sets. It is robust to binding site variations and successfully screens unseen RNA riboswitches in a 20,000-compound in-vitro microarray, with a mean enrichment factor of 2.93 at 1%. This marks the first experimentally validated success of structure-based deep learning for RNA VS.

Only a small fraction of RNA codes for proteins, while ncRNA is now known to play critical roles in a wide range of biological processes[1]. For instance, about 2000 genes code for micro-RNAs, which in turn affect the expression of 60% of genes[2]. Despite this ubiquity, the first RNA-targeting drug, *risdipalm*, was only recently approved by the FDA[3], and nearly all commercially available small molecule therapies still target proteins. The ability to target RNA would drastically increase the size of the druggable space, and propose an alternative for overused protein targets in scenarios where they are insufficient. For instance, lncRNAs could represent interesting therapeutic targets in oncology[4], for which protein targets might be too specialized. RNA targets also represent a therapeutic avenue in pathologies where protein targets are absent, such as triple-negative breast cancer[5]. In this context, RNA is increasingly recognized as a promising family of targets for the development of novel small molecule therapeutics[6–9], highlighting the need for efficient tools for RNA drug discovery[6].

Here, we focus on structure-based approaches which rely on the knowledge of the 3D structure of the RNA of interest to search for candidate compounds. A major advantage of this methodology is to allow for the discovery of binding modes beyond those already known and achieve high specificity[10]. Recently, many binding affinity prediction methods based on machine learning[11–19] were introduced, challenging traditional ones and unlocking state-of-the-art performance. Despite the widely recognized potential for RNA small-molecule therapeutics, the development of computational tools for identifying promising RNA ligands has faced substantial roadblocks compared to computational protein-targeting small molecule discovery[20]. A major reason for this lag is the paucity of available data of RNA-small molecule interactions and affinity measurements. Whereas the RCSB-PDB database[21] contains hundreds of thousands of experimental protein structures, and now hundreds of millions predicted structures from AlphaFold[22], only a few thousand are available for RNA. Similarly,

[1]School of Computer Science, McGill University, Montréal, QC, Canada. [2]Universidad Nacional de Colombia - Sede Bogotá - Facultad de Ingeniería - Depto. de Ingeniería de Sistemas e Industrial, Bogotá, Colombia. [3]LIX, Ecole Polytechnique, IP Paris, France. [4]Mines Paris, PSL Research University, CBIO-Center of Computational Biology, Paris, France. [5]Institut Curie, PSL Research University, Paris, France. [6]INSERM, Paris, France. [7]Max Planck Institute of Biochemistry, Martinsried, Germany. [8]Center for AI in Protein Dynamics, Vanderbilt University, Nashville, TN, USA. [9]Department of Molecular Physiology and Biophysics, Vanderbilt University, Nashville, TN, USA. [10]These authors contributed equally: Juan G. Carvajal-Patiño, Vincent Mallet. [11]These authors jointly supervised this work: Carlos Oliver, Jérôme Waldispühl. ✉e-mail: carlos.oliver@vanderbilt.edu; jerome.waldispuhl@mcgill.ca

databases of protein-ligand interactions such as PDBBind[23] have been widely used for affinity prediction models and contain tens of thousands of experimentally measured ligand interactions for proteins and only around 100 for RNA. In addition to the paucity of data, the uniqueness of biophysical phenomena that underpin RNA folding make it difficult to translate approaches developed for proteins into the RNA domain[24].

In recent years, several computational methods for RNA drug discovery confronted these challenges. Two major strands have emerged: docking-based and direct scoring approaches. For a given pocket-ligand pair, docking approaches estimate the binding affinity by searching over possible poses of a ligand, iteratively scoring different poses while minimizing an energy function. Widely used nucleic-acid-specific docking tools include rDock[25], DOCK 6[26], and RLDOCK[27]. To enhance the docking procedure, alternative pose scoring functions have been proposed such as AnnapuRNA[28], LigandRNA[29], and very recently FingeRNAt[30], and RLaffinity[31]. These methods offer higher confidence screening results but come at the cost of high computational demands: the time needed to search over poses in a single binding site-ligand pair with a docking software is on the order of minutes. Efforts have been made in accelerating docking. First, the ZINC20[32] introduced a large pre-processed ligand database, allowing users to skip some ligand preparation steps. In addition, some methods were proposed to deploy computations on a GPU, and manage to screen compounds in approximately 0.1s per compound, per GPU, without sacrificing accuracy[33-35]. Finally, some protocols tune docking approximation parameters for a specific pocket on a subset of ligands, enabling a faster, less accurate screening running in approximately 0.8s per compound per core.

Instead of undergoing the expensive docking procedure, a few direct scoring methods have been proposed to estimate the binding probability without pose searching, typically using data-driven strategies. Tools such as InfoRNA[36] and InfoRNA2.0[36] score pocket-ligand pairs directly according to the secondary structure similarity to a fixed database of experimentally determined RNA-ligand complexes. Our previous contribution RNAmigos[37] aimed to eliminate the need of a fixed ligand library and made the first attempt at integrating the tertiary structure of the binding site into deep learning assisted compound screening, showing the potential of graph representations based on the Leontis-Westhof nomenclature of base pairs in 3D[38], and outperforming InfoRNA2.0 at native ligand recovery. However, at present, no structure-based RNA virtual screening tool combines the accuracy of docking with the scalability of direct scoring methods.

In this paper, we introduce a structure-based method for virtual screening on RNA that is competitive with molecular docking in a fraction of the time, opening the door to large-scale target-based RNA drug discovery. Whilst our previous work provided insights on RNA binding sites representation as graphs[37], the paucity of RNA small molecule binding data impeded the performance and generalization of this approach, hindering its use in real drug discovery pipelines. We address this data paucity by generating a large database of docking scores, together with unsupervised data, as data augmentations. Moreover, we improve our RNA binding site graph representations and propose model designs that open the door to enhanced neural networks and pretraining techniques. Despite running in seconds instead of hours, the proposed approach retrieves higher enrichment factors to molecular docking (top 2.8% vs 4.1% of the candidate ligand list). Moreover, using the tool and docking jointly, we are able to further enhance docking virtual screening enrichments by cutting computation costs four-fold while improving the predicted rank of actives from 4.1% to 1.0%. This strong performance is found to be maintained over a structurally diverse and unseen set of targets and compounds, and upon binding site perturbation. Finally, we achieve strong enrichment factors and diverse retrieval of diverse leads on an independent experimentally derived ligand screen of 20,000 compounds which constitutes the first evidence of success for structure-based deep learning assisted virtual screening for RNA.

## Results
### Overview
`RNAmigos2` is designed to enable rapid screening of ligand libraries for binders using a query RNA structure. Our pipeline, illustrated in Fig. 1, takes as input a candidate binding site structure (as full 3D or base pairing network) and a list of compounds to screen. The tool then returns a score for each compound reflecting binding likelihood.

`RNAmigos2` model uses an encoder-decoder framework, with two encoders and two decoders, which are each trained on separate data sources. The two encoders are respectively mapping the input RNA binding sites and small molecules into embeddings. The RNA 3D structure is represented as a graph, called 2.5D graph, that encodes all canonical (Watson-Crick and Wobble) and non-canonical base pair interactions occurring in the structure[39]. This representation allows us to capture key features of the RNA 3D architecture with a discrete mathematical object[38,40] that suits machine learning frameworks and was shown to be a useful biological prior for RNA chemoinformatics applications[37]. The RNA encoder takes the 2.5D graph as input and learns to generate an RNA representation using self-supervised training schemes[41] on all available non redundant RNA substructures. Ligands are represented as molecular graphs. The ligand encoder learns a neural representation for ligands, using a variational auto-encoder model proposed in[42,43] and trained on a large dataset of chemical compounds[44].

To train decoders, we extract 1740 RNA-ligand complexes from the PDB[45] and group them in 436 clusters of similar binding sites we identified using RMAlign[46] with a similarity threshold of 0.75. This approach represents a rigorous structure-based split for RNA drug-target association prediction. Our first decoder (`Compat`) is trained as a binary classifier to distinguish between the native ligand of a binding site and decoys. In addition, to synthetically augment the limited number and drug-likeness of PDB compounds, we perform a large-scale docking experiment by docking 500 drug-like ChEMBl chemical compounds on our 1740 binding sites. Our second decoder is trained to predict binding affinity (`Aff`) using the docking data. Given a binding site and a list of ligands, we encode all objects and use our joint decoders to predict a compatibility score that can be used for virtual screening. In what follows, we measure performance of a model by its ability to assign a high score to active compounds against a pool of inactive (decoy) compounds.

### `RNAmigos2` is competitive with docking software and generalizes to new targets
In this work, we propose several enhancements to the original `RNA-migos` encoder model: we expand the dataset and training regimes, represent binding sites with directed graphs, and update the model architecture and pretraining strategy. Following the validation protocol in[37] and our ligand filtering, we show that the proposed enhancements yield a significant performance gain of over 25% AuROC, achieving a new state-of-the-art on this task. The detailed results are presented in Supplementary Section A.2.

We now turn to an evaluation of our results using a larger and drug-like decoy set. The new decoy set is built querying the ChEMBL database[47] such that: (i) at most 100 compounds have a physicochemical similarity $\geq 0.6$ with the native ligand for each binding, (ii) the set has a high diversity determined by the MinMaxPicker algorithm (Ref. 48), and (iii) all compounds pass a drug-like filter, keeping compounds with a molecular weight below 400, at least one ring, less than five rotatable bonds and H-bond donors, less than ten H-bond acceptors and logp values below five[49]. The active compound is defined as the one found co-crystallized in the PDB with each binding site, also termed the native ligand.

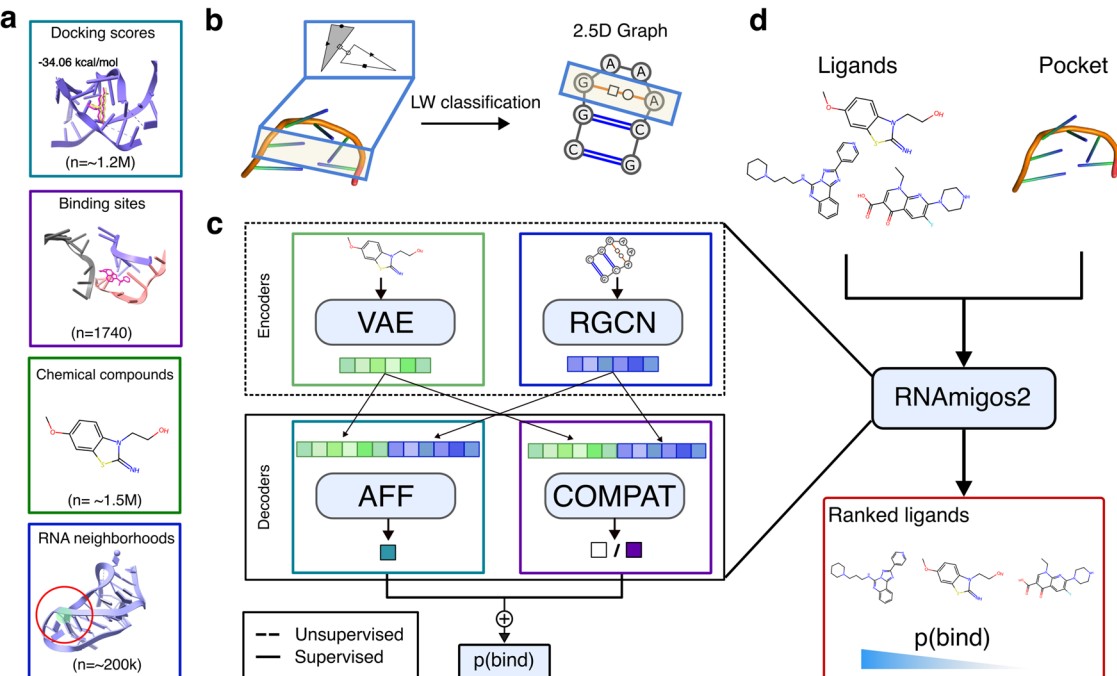

**Fig. 1 | Overview of RNAmigos2 compound screening pipeline. a** Datasets used in this paper. Beyond the supervised data available in the form of 1740 deposited structures of a ligand and its binding site, we use 1.3M molecular docking values as synthetic data and unlabeled datasets of 1.5M chemical compounds and 200k RNA residues neighborhoods. **b** Representation of RNA structure as a 2.5D graph. Each binding site is encoded using Leontis-Westhof classification[38] and converted into an edge-attributed directed graph. **c** Models to learn RNA-ligand interactions. We use an RNA and a ligand encoder, pre-trained in an unsupervised way. We then decode these representations to predict binding affinity (Aff) trained using our synthetic data; or a compatibility score (Compat) trained using deposited complexes. **d** RNAmigos2 integrates these models to screen input ligands libraries, sorting them by their predicted probability of binding.

On the binding site side, we enforce non-redundancy by computing the 3D alignment-based RMscore[46] over all pairs of binding sites in our dataset and perform a hierarchical agglomerative clustering with a cutoff of 0.75, resulting in 436 clusters of binding sites. The resulting binding site groups represent similar binding sites or different instances of the same binding site with potentially different ligands. Native ligands of binding sites belonging to the same group are considered as positives for this group. Then, we performed a data splitting on the groups, ensuring no structural data leakage exists between our training set of 367 groups of binding sites and our test set of 69 binding sites.

On this new dataset, we train models using our two settings: Aff and Compat, comparing performance to rDock on the test set. We find that Aff does indeed correlate well with rDock intermolecular energy terms (Spearman correlation of 0.75) on unseen binding sites, as well as shows a clear separation between native ligands and decoys with respect to both docking and Aff scores (see Fig. 2a). Applying our models in a VS setting (Fig. 2c) we achieve a mean AuROC score of 0.844 and 0.939 using Compat and Aff models respectively. We emphasize the high performance of the Aff model that was trained solely on simulated data. It is worth noting that such high values are common in VS, as libraries of hundreds of thousands of compounds are screened but only a few thousands of compounds in the top percentiles can be sent to wet-lab assays. On the same task, rDock slightly outperformed our models with a mean value of 0.959. This result is expected given the intensive (and CPU-time costly) optimization process executed by rDock in identifying the best poses. We underline however that the runtime of our method is less than five seconds on a single machine, whereas the docking results were obtained after approximately 8 CPU hours per binding site.

Interestingly, we find that the errors in VS are aligned for each binding site for our different models (see Fig. 2d). While as expected, rDock and Aff errors align, the Compat trained on an independent

data source has complementary error modes. This complementarity might originate from force field priors being too strong to accurately fit the data. This suggests a simple strategy for mitigating their individual errors, and we propose to ensemble each model's results by taking the max of their ranks to create a model coined as Mixed. Our averaging is reminiscent of Bayesian fitting, with a likelihood term and a physical prior term.

The Mixed scores now have a lower correlation of 0.55 with docking values (Fig. 2b), but we see that this correction enables to effectively recover active compounds that the docking program was not able to identify alone. We compare the VS performance of this mixing with an ensemble of two models trained on the same data source, and report AuROCs over 3 independent seeds for all pairwise combinations in Supplementary Table 1. We reach a performance of 0.972 AuROC for the Mixed to be compared to 0.897 and 0.942, $n = 65$, for the ensembled versions of our single modality models. This performance gap proves the synergistic effect of our models. It also enables us to significantly outperform rDock (0.959, Mann-Whitney U test p-value of 0.026). Hence, the performance boost obtained by correcting docking scores with experimental data exceeds the decrease resulting from our imperfect docking surrogate model, which establishes a new state-of-the-art machine-learning based RNA virtual screening tool.

Next, we coin our best performing configuration, Mixed as RNAmigos2 for simplicity and evaluate it in a larger benchmark of RNA virtual screening tools and study the generalizability of our predictions. We compare the resulting model with several other VS tools, namely RLDOCK[27], AutoDock+Vina[50], Annapurna[28], DOCK 6,[51] rDock[25] and RNAmigos1[37] on a virtual screen against ChEMBL compounds. As presented in Fig. 3a, our method outperforms all methods with a wide margin and a smaller margin for rDock as seen in Fig. 2c. We report minimal variance in our models' performances by training them over three random seeds and include these values in the first column of

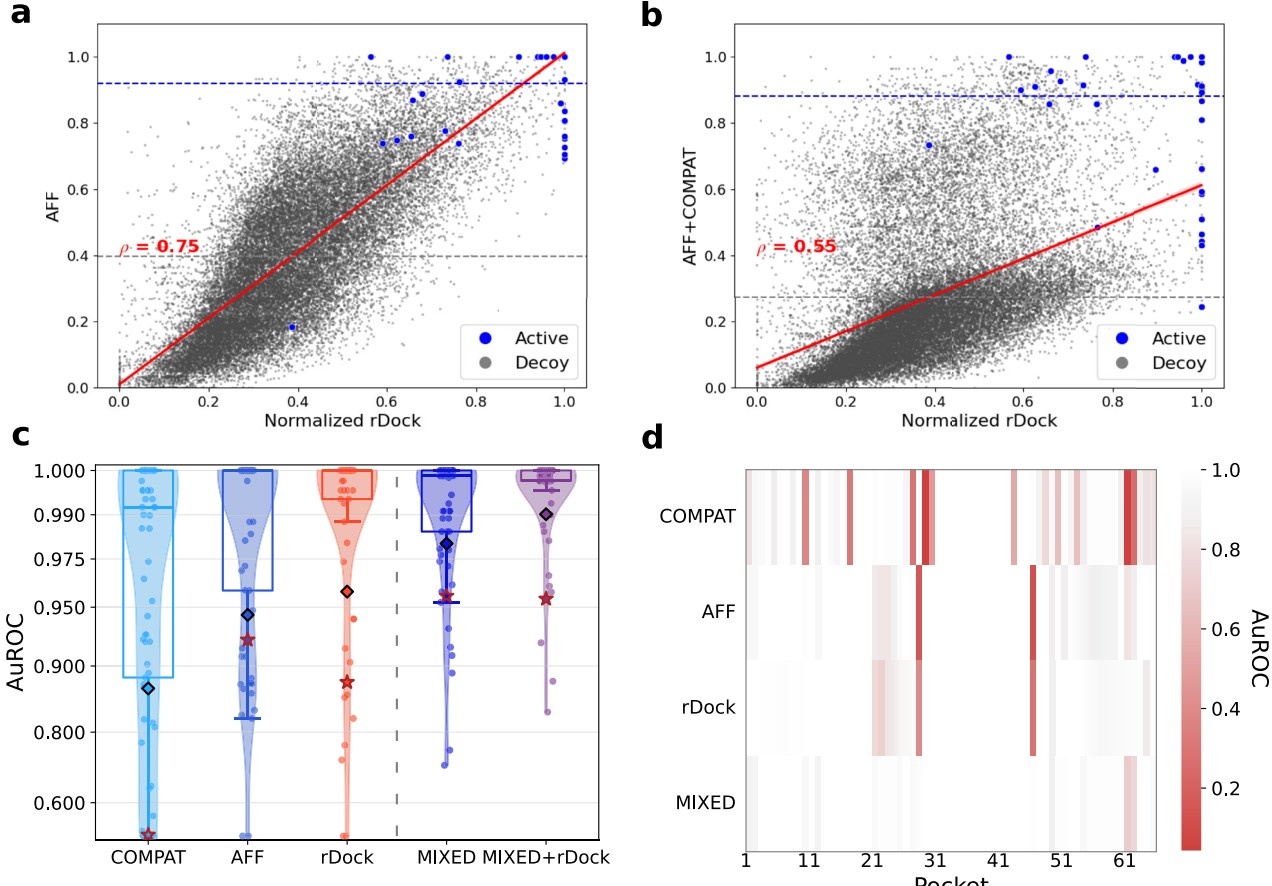

**Fig. 2 | RNAmigos2 model ensembling benchmark. a, b** Relationship between Aff (**a**) and Mixed (**b**) predictions against rDock scores for decoys and native ligands in the test set. Horizontal lines indicate the mean of predicted scores for native and decoy ligands. **c** Distribution of the AuROC performance of different virtual screening models over test binding sites. Diamonds represent the mean performance and stars represent performance with a shuffled target ($n = 65$), boxes represent quartiles. We compare models trained under our two settings with rDock (left of the dashed lines), an ensemble of these models (Mixed) and the average of Mixed and rDock. **d** Description of the performance of the tools split by binding sites. Each line on this plot corresponds to a point in (**c**). This shows that the successful predictions of different methods are complementary and illustrates their synergy in a mixed model.

Supplementary Table 1. Next, we look at the generalization capacity of `RNAmigos2` when considering the novelty of test set pockets and ligands with respect to the training data. For each binding site in the testing set we report the AuROC versus the structural similarity (RM-score[46]) to the most similar binding site in the training set (see Fig. 3b). We see that the model performs consistently well at ranges of structural redundancy from 0.3 up to the threshold value of 0.75.

Finally, we assess our model specificity and robustness. Specificity is known to be hard for drug-target interaction prediction. Indeed, decoy compounds tend to have a chemical bias, such that active compounds can be found by a simple QSAR model. An interesting swapping experiment is proposed in Volkov et al.[52]. It consists in shuffling target-ligands pairings, so that compound affinities for a target are associated with another target chosen at random. The authors manage to retain high enrichments for models trained of shuffled pairs, disproving their specificity. While some enrichment is expected based on general properties of active chemical compounds, a high specificity model should display as large a performance gap as possible on shuffled targets. We visualize the results of this experiment on Fig. 2; by denoting the swapped performance with a red star. It can be seen that our `Aff` model has a small gap of 2.0 AuROC points, while rDock has a gap of 7.8 points. Thanks to a margin loss term correcting for this bias, we report a high pocket specificity in the `Compat` setting, where swapping induces a 41.7 point performance drop. This allows our `RNAmigos2` model to show target specificity, where swapping results in a drop from 98.1 to 95.6 AuROC.

Another key aspect of virtual screening is its robustness to the definition of the pocket. Indeed, current state-of-the-art binding site detection methods[53] retrieve 70% of the correct residues, and potentially miss the exact localization of the pocket. We conduct an experiment to assess the robustness of our tool with regard to imperfect pockets. We implement algorithms for sampling perturbed binding sites, precisely described in Supplementary Section E.2. For each pocket $p$, we first expand it with a breadth-first search of $h$ hops around the original binding site, yielding the expanded pocket $p_h$. Then, either we sample nodes in $p_h$, yielding a wrong sampling in the right average localization (*noised*), or around a seed node at the boundary of $p_h$ (*shifted*), mimicking errors in the binding site localization. The number of sampled nodes is computed as a fraction of the size of $p$, ranging from 0.5 to 1.3, and the sampling is replicated ten times. We present results in Fig. 3c, d.

Our methods display some sensitivity to those perturbations, in line with the swapping experiments results. However, when sampling nodes in the close vicinity of the pocket (*noised*, $h = 1$), we are able to retain most performance with about 0.7% AuROC drop. When growing $h$, the pocket becomes more scattered, smoothly decreasing our results to reach a 2% drop. When perturbing the localization, the drop ranges from 1.8 to 3%, which is more significant, but remains a strong signal considering that the most disrupted pockets barely overlap with the original pocket. At the highest level of perturbation, the performance can be worse than one obtained over swapped pockets, which can be explained by perturbed pockets that do not correspond to a

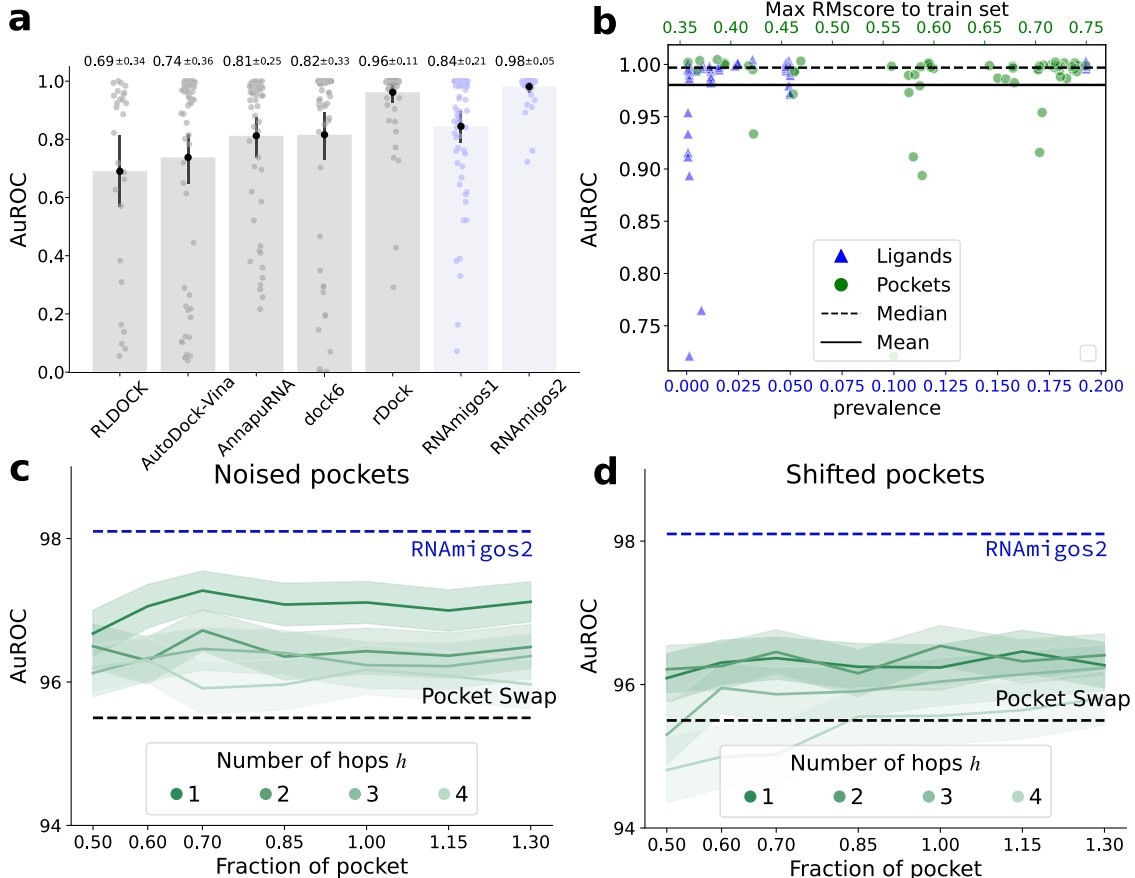

**Fig. 3 | Docking tool and binding site perturbation benchmark. a** Virtual screening benchmark of RNAmigos1 and 2 vs state of the art docking tools on ChEMBL compounds. Bars represent 95% confidence intervals, $n = 69$. Note, RLDOCK produced errors on 42 binding sites. We only report its performance for 27 binding sites, which we denote with RLDOCK*. **b** Depiction of the performance of RNAmigos2 split by similarity of the test binding site to the train set. This similarity is assessed with RMScores for the binding sites, and Tanimoto similarity of chemical fingerprints for ligands. **c, d** Mean and standard error of the AuROC for different perturbations ($n = 10$ independent runs of pocket noising). For each pocket p, we first expand it with h hops to get ph, and either sample nodes in ph (noised), or around a seed node at the boundary of ph (shifted). The number of nodes we sample is computed as a fraction of the size of p. The sampling is replicated ten times. Unperturbed performance is shown in blue.

pocket anymore. Finally, we observe a remarkably stable behavior with regards to the fraction of the pocket $f$, which is a very promising result for synergic usage with binding site detection models. To explain this stability, remember that we expand our binding sites with context before embedding it. Hence, we believe that our model implicitly learns to detect the exact binding site from an approximate location, explaining its robustness.

**RNAmigos2 predictions consistently boost virtual screening efficiency**
Our model outperforms docking accuracy in a fraction of the time. However, since rDock displays a strong performance, we also investigate mixing the RNAmigos2 model with docking, and show it enhances performance from 98.1% to 99.0%, halving the error of these models. Interestingly, this also increases the pocket specificity of our model. We refer to this more accurate, but more expansive strategy, as RNAmigos++, and plot its performance in Fig. 2c.

In practice, we have a limited time budget. Existing alternatives can bring this run time to 0.1 GPU seconds/compound, but rely on the availability of a GPU, or to 0.8s core second/compound at the cost of a less accurate result. We found docking with rDock to take approximately 65 core seconds/compound. Hence, we cannot afford to scan large scale databases with docking methods, which introduces a trade-off in virtual screening between quality of the scoring and the amount of computational time expended. In contrast, RNAmigos2 needs 5.8

core ms/compound on raw smiles. We even reach 1.6 core ms/compound on precomputed ligand graphs (a one-time overhead cost), which can be useful for projects involving several targets. This scoring can be considered negligible compared to docking run time. Hence, in a drug design setting and under a compute budget of one core day, conventional docking enables the screening of about ~ 1400 compounds while RNAmigos2 enables screening over 15.4M compounds.

To leverage the performance gain obtained with RNAmigos++ while keeping computations tractable, we propose to quickly presort compounds according to our RNAmigos2 score, and then sort the best scoring ones again with RNAmigos++ (See Fig. 4a). We expect that presorting the compounds will help the docking tool to review the most promising ligands first and thus use the allocated compute time more efficiently. We compare this hybrid strategy with the sole use of docking and with RNAmigos2 in Fig. 4b: we report VS performance of the different methods for different time budgets ($x$-axis). For a time budget lower than the maximum, only the first compounds are sorted using our methods, whilst remaining ones are kept in their initial order at the end of the list. Introducing the *efficiency* of a VS method as the area between the curves of AuROC through time for this method and rDock (further developed in Methods), the *efficiency* of our hybrid method is shown in Fig. 4c.

This benchmark highlights that our methods obtain excellent results much faster than with a docking-only approach. Using RNAmigos2 in isolation, we get almost the same AuROC as rDock in

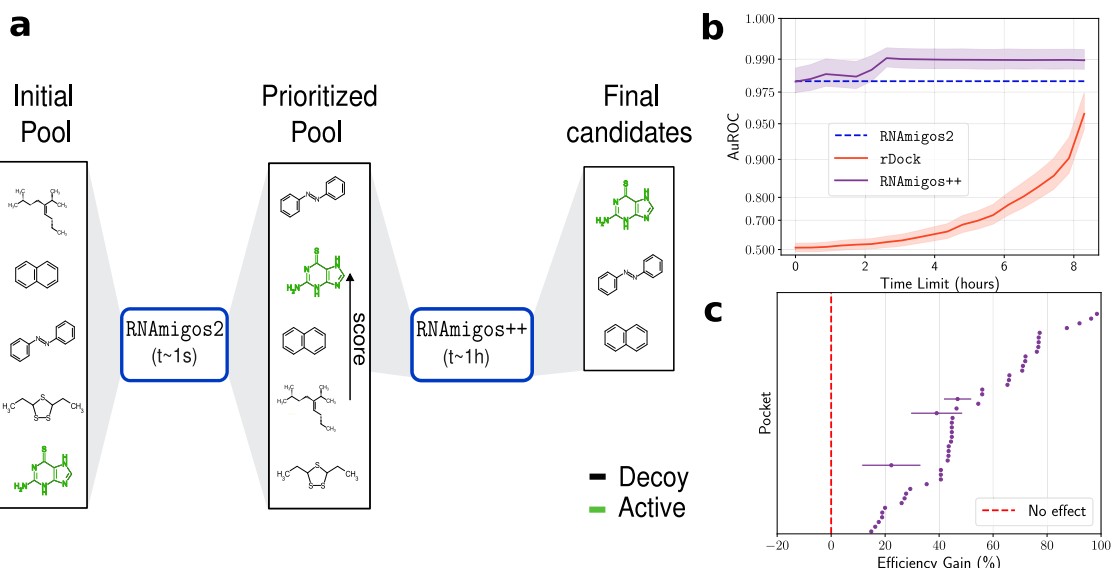

**Fig. 4 | RNAmigos2-assisted virtual screen efficiency. a** Illustration of our hybrid compound screening strategy. First, we sort the compounds using RNAmigos2 with a high throughput. Then, rDock is run only on the best scoring compounds and used to replace our docking surrogate score. **b** AuROC of different models, averaged over binding sites, as a function of time budget spent. Standard error of AuROC results obtained with ten shuffled replicates of the initial order of the list.

Note the logarithmic scale on the *y*-axis, that gives the impression of acceleration for rDock screening results. It is actually a linear interpolation between random and final performance. **c** *Efficiency* metric split by binding sites. Each line corresponds to a test set pocket, error bars represent standard error of efficiency gain across 10 initializations of the virtual screen.

11,000x less time. Screening all compounds with both tools cuts rDock error by a factor four. Crucially, the boost of performance with `RNA-migos++` over the `RNAmigos2` model is achieved after screening only one fourth of the compounds. Finally, we show that we improve the performance for almost all binding sites, which indicates that our protocol never results in a waste of time. These results suggest that our method could allow the screening of larger libraries, potentially those coming from ligand generative models[54].

### `RNAmigos2` identifies riboswitch ligands in large-scale in vitro assay

We now aim to assess the performance of `RNAmigos2` in a realistic large-scale scenario with in vitro validation. In the previous experiments, we established ground truth using co-crystallized ligands from the PDB as the active (native) compounds. Here, we use a recent large-scale in vitro screen of approximately 20k compounds against 36 nucleic acid targets, known as ROBIN[55]. As in preceding experiments, we seek to identify the active compounds for each target which in this case are the range of 163 to 197. To make predictions using `RNAmigos2`, we identify all RNA targets in the ROBIN database which have a perfect sequence identity to an RNA chain in the PDB and pass the corresponding 3D structures as input to the model. Importantly, we ensure that all RNAs collected were not present in our training or validation sets, making this a blind testing scenario. As `RNAmigos2` requires a binding site as input, we use the native ligand in the PDB as before to define the input region. This results in four binding sites: the TPP, ZTP, SAM_ll and PreQ1 riboswitches, (PDB ids: `2gdi`, `5btp`, `2qwy` and `3fu2`), visualized in Fig. 5a.

We computed the `RNAmigos2` scores of all 24,572 assayed molecules for all pockets in about two CPU minutes. Our first finding is that `RNAmigos2` yields positive enrichments of active compounds in all four targets (AuROC reaching 0.66 and enrichment factors up to 5.09), significantly distinguishing decoys from actives (Fig. 5b, Mann-Whitney U Test scores range $3.73 \times 10^{-15} - 1.61 \times 10^{-6}$) and despite both the compounds and targets being unseen during training. Remarkably, performance of `RNAmigos2` on its own is on-par with rDock while running in a fraction of time (2 core minutes instead of

1000 core hours ; see Fig. 5, and Supplementary Table 3 for full results). Combining our predictions with rDock (`RNAmigos2++`), though more time-consuming, achieves higher accuracies than either tool on its own in 3 of 4 pockets, and always improves screening efficiency (Fig. 5c).

Beyond AuROC and enrichment factors, we report that using `RNAmigos2` augments the chemical diversity of retrieved actives. Retrieving diverse actives is a key feature of virtual screening methods, providing more opportunities for hit optimization as well as lowering the risk of systematic failure on later tests[56]. Thus, two actives from different chemical scaffolds can be more valuable than five related active compounds, which is not measured by AuROC. In Supplementary Fig. 7, we show actives retrieved by different methods. It is evident that our `Aff` model focuses on main clusters, which represent generic RNA binders and allows this model to get high AuROC scores. This also holds partly true of rDock, but is in contrast with our `Compat` model that seems to focus on on binders specific to a pocket, in line with its margin loss training. This equips `RNAmigos2` with enhanced diversity compared to rDock, as measured by mean Tanimoto distance to the closest neighbor, as well as by the Hopkins statistic[57]. More importantly, we find in Supplementary Fig. 3 that the distribution of selected actives by `RNAmigos2` and rDock show notable Earth Mover's Distance[58,59] shifts of (around 0.3) suggesting we explore different areas of chemical space. This is illustrated on ZTP for instance, where the left cluster is only found by rDock but the bottom right part is retrieved only by our tool. This is an opportunity for `RNAmigos++` that displays the highest diversity on all targets. We believe having several fast scoring functions with complementary sensitivity is an exciting avenue for obtaining truly diverse and high-accuracy virtual screening protocols.

## Discussion
In this work, we address the timely question of in silico RNA drug design. In contrast to proteins for which we already accumulated a lot of structural data, the current sparsity of reliable three-dimensional RNA structures prevents the application of machine learning paradigms for RNA drug discovery tasks. The development of dedicated

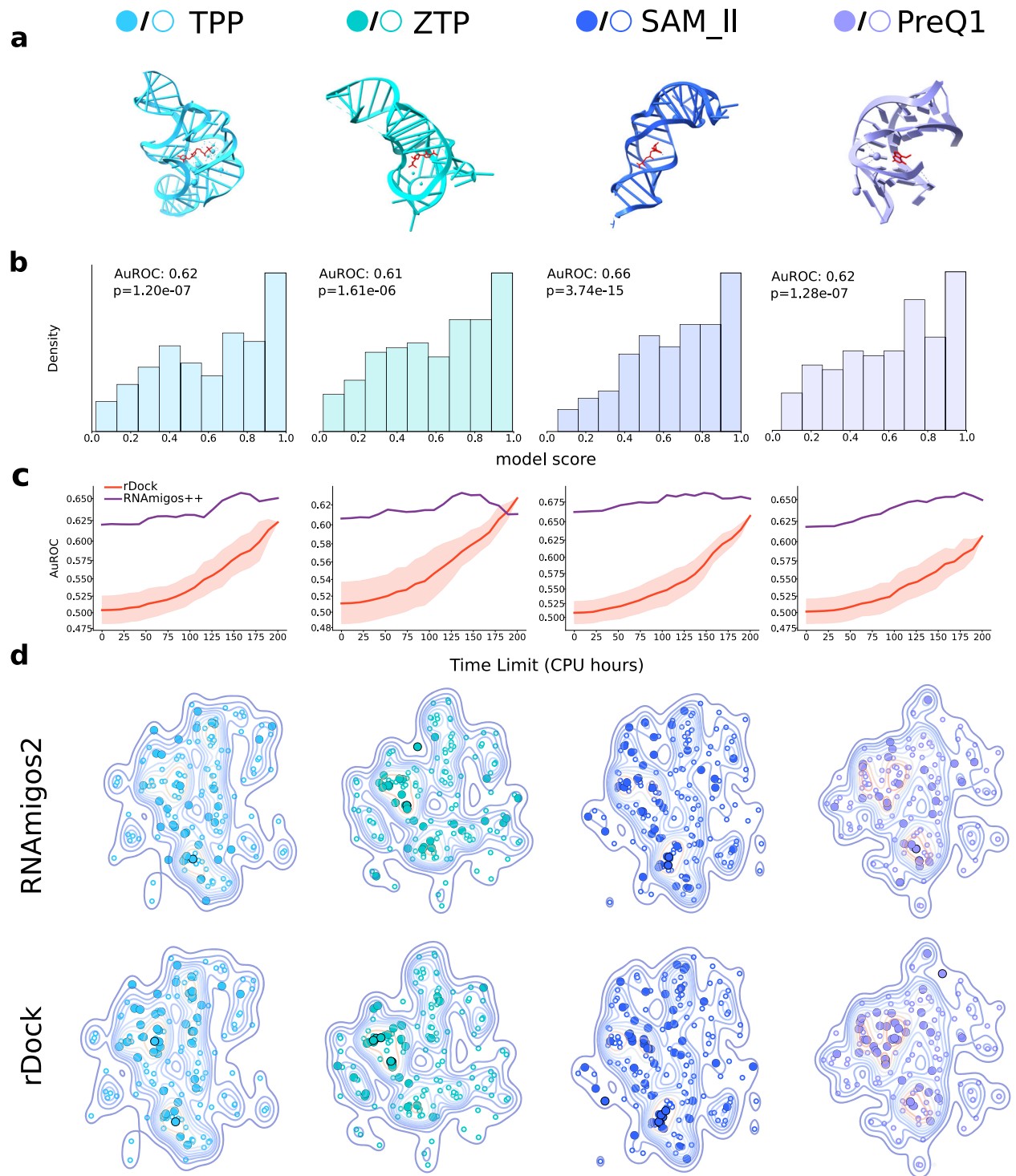

**Fig. 5 | Zero-shot screen of in vitro binding data. a** Visualization of the structure of the riboswitches targets in the ROBIN dataset used in this experiment. **b** RNAmigos2 ranking of active compounds in the ROBIN database for each target. P-values reported are from the Mann-Whitney U test for comparing distributions of ranks of active (+) and inactive (−) compounds (TPP $n^+ = 163$, $n^- = 24340$, ZTP $n^+ = 171$, $n^- = 18585$, SAM $n^+ = 167$, $n^- = 20473$, PreQ1 $n^+ = 167$, $n^- = 24336$). **c** Time-limited benchmark on each ROBIN target using RNAmigos2 to prioritize compounds for rDock. Error bar represents standard error across 10 random initializations of the virtual screen. **d** Chemical compounds selected (best 10%) by different methods, shown in the tSNE space of ROBIN active compounds. Large, filled circles correspond to selected actives and hollowed ones are non-selected actives. A kernel density estimate is shown beneath the points.

frameworks and methods are therefore essential to address this bottleneck.

Specifically, we complement the available RNA-ligand complexes structures by using RNA docking as a data augmentation strategy and incorporate domain-specific unsupervised pre-training techniques. Whilst our model runs more than ten thousand times faster than docking, it replicates its performance for diverse targets. This result is sensitive to pocket identity, but robust enough to pocket perturbation to be used in conjunction with modern pocket mining algorithms[53,60]. Moreover, by combining our model with actual docking scores for the top-scoring compounds, we manage to cut docking error by four, in one fourth of the time. Finally, we

establish our tool's performance on an independent large-scale (25k compounds) in vitro binding screen, and show that it provides enrichment factors at 1% of 2.93 in two CPU core minutes. Together, these results establish `RNAmigos2` as the new state of the art in structure-based RNA virtual screening.

By publicly releasing all our datasets, source code and model weights, we hope to spark a community effort in this important direction. Limitations to our approach include the need for pre-defined binding sites for which integrations with binding site predictors will need to be developed, as well as for the modeling of binding site flexibility[61]. An interesting direction for future work is to investigate the use of other docking tools to train other models, resulting in fast surrogates with potentially different error modes resulting from different scoring functions. We envision tools such as `RNAmigos2` will play a synergistic role with rapidly emerging RNA-centric technologies for molecular design[62] and newly released AlphaFold3[63] which supports nucleic acids to pave the way for the next generation of RNA drug discovery. Notably, our methods have the distinct advantage to enable structure-based virtual screening for RNA with only low-resolution structural data at hand such as base pair interactions. In light of the daunting number of potential RNA targets, such feature could prove itself a major asset to mine entire genomes and fully embrace the era of RNA therapeutics[64].

## Methods

### Constructing and representing a dataset of binding sites

To construct our dataset of binding sites, we download all RNA structures from the Protein Data Bank (PDB)[21] containing RNA and at least one ligand. Then, we extract RNA binding sites as RNA residues with an atom at most 10Å from the ligand, and filter out resulting binding sites with less than five RNA residues or where more than 40% of the total residues in the 10Å zone of the ligand come from protein chains, following. We additionally remove complexes with ligands that did not follow rules introduced in Hariboss[65–67] (e.g. removing ions, crystallization artefacts, etc.). The final dataset consists of **1740 binding sites** associated with **264 unique ligands**.

We then use RNAGlib[68] to obtain a representation of the 3D structure of our binding sites in the form of 2.5D graphs. The nodes of these graphs represent nucleotides and contain a one-hot encoding of the nucleotide type. Each node also contains a 640-dimensional feature vector from the RNA-FM language model[69] which was trained on a large corpus of RNA sequences and provides a strong evolutionary signal. Edges represent interactions between nucleotides, split in interaction types that follow the Leontis-Westhof classification[39]. Briefly, the Leontis-Westhof classification is a 12-way discrete categorization of base pairing interactions according to orientation of interacting bases[39]. This classification is known to capture salient geometric characteristics of the RNA, and thanks to its graph-based nature, it is highly amenable to geometric deep learning frameworks[39,41]. An undirected 2.5D graph representation was shown to give better performance[37] than traditional representations for machine learning. In this work, we additionally use directed edges, based on base-pair orientation, to disambiguate sequence orientation and asymmetrical non-canonical interactions. We expanded our graphs by performing a breadth-first expansion of depth four, to add context nodes around the binding sites.

### Unsupervised datasets of RNA structures and chemical compounds

We use two additional sources of unsupervised data to pretrain our networks. We use all RNA structures in the non redundant dataset proposed in[70], represented as directed 2.5D graphs, and extract the 2-hop neighborhoods around each residue, resulting in over 200,000 small RNA graphs. We additionally use a curated dataset of ~1.5M ZINC[44] compounds, introduced in[71] as reasonable drug-like

compounds. These molecular compounds are represented as molecular graphs with nodes containing atom types, charge and aromaticity and edges with different chemical interactions.

### Docking dataset construction

Our current database only includes co-crystallized ligands, that are thus positive binding examples. We do not have information about the affinity between a binding site and the native ligands of other binding sites, which are thus assumed to be negative pairs. Moreover, we would like to have more affinity data about drug-like compounds, since PDB compounds are not necessarily drugs (cofactors for instance). To generate more data, we use molecular docking as a source of synthetic data.

Therefore, we curate a list of drug-like compounds from ChEMBL[47]. To do so, we use a drug-like filter and keep compounds with a molecular weight below 400, at least one ring, less than five rotatable bonds and H-bond donors, less than ten H-bond acceptors and logp values below five[49]. We complement this procedure with the MaxMin algorithm based on Tanimoto distance between fingerprints, using RDKit[48], for picking a subset of 500 diverse compounds, ensuring that up to 100 compounds have a Tanimoto Coefficient greater than 0.6 with the native ligand for each binding site.

To perform our molecular docking, we compared three candidate tools, rDock[25], DOCK6[51] and AutoDock Vina[50] which were identified in a recent review of nucleic-acid docking tools[72]. rDock was developed for NA-ligand docking, DOCK 6 is an extended version of DOCK 5 specifically optimized for RNA and AutoDock Vina was used without any specific optimization for NAs. In addition, we used AnnapuRNA[28], a knowledge-based scoring function, as an independent referee to compare these candidate software in terms of native pose identification. We first establish our docking procedure and compare our candidate docking programs using self and cross docking experiments. In self-docking experiments, we evaluate programs' ability to predict the binding modes inside the native macromolecule by comparing the generated ligand poses to the crystal structure. In cross-docking experiments, we evaluate the ability of docking tools to retrieve actives among decoys. An in-depth explanation of these experiments is available in Supplementary section D. Based on our results, we choose to use rDock as our docking program. We used it to dock the ligand sets of 264 native binders of the PDB and 500 ChEMBL compounds in our 1740 binding sites, resulting in over 1.3 million values. On the test set, the docking scores were used to obtain virtual screening performance of rDock for benchmarking purposes. The ones obtained for the train set binding sites are used as a data augmentation strategy.

### Data splitting

We compute the RMscore[46] of all pairs of binding sites in our dataset, and report their distribution in Supplementary Fig. 1a. As can be seen, while most of the binding sites are distinct, there exists a proportion of binding sites that are highly similar. Hence, we then perform a hierarchical agglomerative clustering with a cutoff of 0.75. This allows us to group similar binding sites together and results in **436 groups** of binding sites.

Those groups can represent similar binding sites, or different instances of the same binding site with different ligands. Data is split based on the groups, ensuring no structural data leakage exists between our training set of 367 groups of binding sites and our test set of 69 binding sites. An MDS representation of our train and test groups is available in Figure 1b, showing that our test points are diverse in the space of binding sites. Training and testing on binding site groups requires caution. To avoid data imbalance, we chose to use only one representative graph per group, and to set all ligands of this group as group actives. While testing and to compute AuROCs, all the group actives are used as positives.

 

## Models

Our model adopts an encoder-decoder framework with two encoders and two decoders. We use a neural network encoder $f_\theta$ based on Relational Graph Convolutional Network (rGCN)[73] layers. Given the set of all relations $\mathcal{R}$, we denote as $\mathcal{N}_r(i)$ the set of neighbors of node $i$ connected by relation $r$. The parameters of this layer $l$ are $r+1$ matrices $W_r^{(l)}$ with the convention that self loops have an index 0. The node representation of node i at layer l, $\mathbf{h}_i^{(l)}$ evolve to the next layer following the equation:

$$\mathbf{h}_i^{(l+1)} = \mathbf{W}_0^{(l)}\mathbf{h}_i^{(l)} + \sum_{r \in \mathcal{R}} \sum_{j \in \mathcal{N}_r(i)} \frac{1}{c_{ir}} \mathbf{W}_r^{(l)} \mathbf{h}_j^{(l)}.$$

where $c_{ir} = |\mathcal{N}_r(i)|$. This update rule aggregates the representations of neighboring nodes based on their relational types and updates the target node's representation accordingly. We apply several rGCN layers to our input, resulting in node embeddings of dimension $d \in \mathbb{N}$ and a pooling step reduces those node embeddings to a single graph-level vector representation $\phi_\mathcal{G} = f_\theta(\mathcal{G})$.

To encode the structure of small molecules, the ligand is represented as a molecular graph $\mathcal{F}$ to serve as input to our model. This graph is then fed into a ligand encoder, denoted as $g_\theta$, resulting in $\phi_\mathcal{F} = g_\theta(\mathcal{F})$. We use the architecture proposed in OptiMol[42], which is a stack of GCN layers (obtained by setting $r = 1$ in the rGCN equation above).

We then concatenate the graphs and ligand embeddings and feed the result to a decoder that outputs real values, so that we can write our overall models as $M(\mathcal{G}, \mathcal{F}) = h_\theta(f_\theta(\mathcal{G}) \bigoplus g_\theta(\mathcal{F})) \in \mathbb{R}$. The specific model architecture and hyper-parameter settings are described in Supplementary section E.1.

## Pretraining

Our encoders are pre-trained to serve as a strong initialization for the representation of the input binding sites and the small molecules. Our RNA encoder pretraining relies on the strategies proposed in[41] which are based on the metric learning framework[74]. Namely, given a similarity metric $k$, the network is trained to minimize

$$\mathcal{L} = ||\langle \phi_{\mathcal{G}_1}, \phi_{\mathcal{G}_2} \rangle - k(\phi_{\mathcal{G}_1}, \phi_{\mathcal{G}_2})||^2.$$

Using directed graphs opens the door to using more powerful similarity functions $k$ than the ones used in RNAmigos1, and hence enhanced pre-training. In this work, the graphs $\mathcal{G}_1$ and $\mathcal{G}_2$ are defined as subgraphs of whole RNAs, expanded from two root nodes. Then, we compute graphlets decomposition of these neighborhoods, and compute a pairwise distance matrix between graphlets of $\mathcal{G}_1$ and $\mathcal{G}_2$, using a graph edit distance adapted for RNA similarity. Finally, the $k$ function uses the Hungarian algorithm[75] to match the nodes following this pairwise distance matrix, and returns the score of the match.

Our ligand encoder, $g_\theta$, is pretrained following OptiMol[42]: in addition to its graph encoding, we encode molecular compounds as a SELFIES[76] string. The encoder is then trained in conjunction with a decoder that aims to reconstruct this string from $\phi_\mathcal{F}$ following a Variational Auto-Encoder (VAE) framework. This decoder is not used in our work, but we used the weights of this encoder as the initial weights of our ligand encoder.

## Training

Our first decoder (`Compat`) is trained to output one for native ligands and zero for decoys, by minimizing a Binary Cross Entropy (BCE) loss over experimental complexes. This problem is very unbalanced because we have many more negative pairs than positive pairs. Hence, at each epoch, we sample one active and one inactive of each group and train only on those two examples per group. In addition to this loss, and in order to increase the specificity of our model, we added a margin loss $\mathcal{L}_m$ that aims to set apart the score of a ligand $\mathcal{F}$ with its active $\mathcal{G}_\mathcal{F}$ from the one of $\mathcal{F}$ with another pocket for which it is inactive

$\widetilde{\mathcal{G}}_\mathcal{F}$. Given a margin $\alpha \in [0, 1]$, this loss can be written as:

$$\mathcal{L}_m = \max(M(\widetilde{\mathcal{G}}_\mathcal{F}, \mathcal{F}) - M(\mathcal{G}_\mathcal{F}, \mathcal{F}) + \alpha, 0).$$

Our second setting (`Aff`), keeps the same model described above, but changes the training procedure so that the prediction target becomes the inter-atomic term of the docking score. However, docking distributions are also unbalanced, with only a few high-scoring compounds that we are most interested in. Thus, we normalize our docking scores by using a quantile transform with 50 bins on each pocket, resulting in values between zero and one. We then train our network to approximate those values using an $\mathcal{L}_2$ loss.

## Inference and validation with virtual screening

Virtual screening (VS) aims to sort a library of compounds such that the active ones are at the top of the list[77]. Typically, a VS is used to screen millions of compounds, as only a few thousands can be sent to wet-lab assays. Metrics for VS quantify the gain in active compounds obtained by selecting compounds using a VS strategy instead of a random selection. The main VS metric are the well known AuROC and Enrichment Factors[78], where the enrichment factor at a cutoff $X$ of a VS method is the fraction of actives versus decoys (non-actives) in the top scoring $X$% relative to the overall active fraction.

Because the number of drug-like compounds available in databases such as Enamine[79] is in the billions, we introduce a novel metric, denoted *efficiency*, to measure the expected performance of a screening after a certain time budget is spent. While performing a virtual screening over a list of compounds, we keep track of the AuROC as a function of time and define the *efficiency* of a search as the area under the time vs. AuROC curve. The value of this metric depends on the initial ordering of the list. For instance, if all active compounds are at the end of the list, they are processed last, yielding zero AuROC until these compounds are reached. To get an expected efficiency, we thus repeat the computation over many shuffled initial orderings. In this manner we can capture the trade-off between time expenditure and the quality of the search results in a single value.

To perform virtual screening with our tool, we simply compute the scores for all of our compounds and sort them based on this score. When mixing our models, we compute scores independently for the `Compat` and the `Aff` and then take the best predicted rank for each compound. To use our tool in conjunction with docking, we quickly presort the compounds using our mixed score and only dock the prioritized compounds. For those compounds, we use the true docking score instead of their approximation by the `Aff`.

## Reporting summary

Further information on research design is available in the Nature Portfolio Reporting Summary linked to this article.

## Data availability

All data generated in this study have been deposited in the Zenodo database under accession code https://doi.org/10.5281/zenodo.14803961. The data are available under the MIT license.

## Code availability

All source code is available on GitHub under repository https://github.com/cgoliver/rnamigos2/. Source code is available under the MIT license. Official release to reproduce results is registered on Zenodo under accession code https://doi.org/10.5281/zenodo.14816240.

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

## Acknowledgements

V.M. was partly supported by the ERC Starting Grant No. 758800 (EXPROTEA). This work was supported by a Génome Québec grant (Genomics Integration Program), a Fonds de recherche du Québec Nature et technologies grant (Team research project), and a NSERC Discovery grant to J.W. The authors would like to thank Nicolas Moitessier and David Hiraki for helpful comments and suggestions.

## Author contributions

J.C.P. built training and testing data, trained models, conducted experiments, and revised the manuscript. V.M. conceived project, designed models and experiments, trained and evaluated models, did the figures and led the writing of the manuscript. D.B. provided technical guidance for dataset construction and reviewed the manuscript. L.F.N.V. provided supervision. C.O. conceived project, designed models and experiments, trained and evaluated models, did the figures, led the writing of the manuscript, and oversaw the project. J.W. conceived project, provided supervision, revised the manuscript, acquired funding, and oversaw the project.

## Competing interests

The authors declare no competing interests.
