## [Transparent Peer Review file · Nature Communications]

RNAmigos2: accelerated structure-based RNA virtual screening with deep graph learning

Corresponding Author: Professor Jerome Waldispuhl

Version 0:

Reviewer comments:

Reviewer #1

(Remarks to the Author)

In this manuscript, the authors proposed a structure-based machine learning pipeline, termed RNAmigos2, for fast and accurate large-scale virtual screening of compounds against RNA targets. The model employs an encoder-decoder framework and leverages graph representations for both RNA and ligands. The RNA and ligand encoders are pre-trained using unsupervised graph learning on extensive RNA and ligand structural datasets. Subsequently, the downstream decoders are trained by utilizing both experimental and synthetic data collected from PDB and docking software to identify native binding ligand from decoys and reproduce the docking scores, respectively. A key feature of the proposed approach is the massive speedup achieved over traditional docking-based approaches while still obtaining higher enrichment performance in RNA-targeted virtual screening. Notably, the authors provide detailed steps on GitHub for reproducing the manuscript's results. This work showcases the great potential of deep-learning techniques in modeling RNA-ligand interactions, particularly in structure-based virtual screening workflows, and can be highly valuable to the scientific community.

Comments:

1. In Section "RNAmigos2 is competitive with docking software and generalizes to new targets". The RLDOCK model provides a nice scoring function for R-L docking, but the model is not designed for VS because RLDOCK contains a ligand intramolecular LJ energy term that cannot be directly compared between different compounds. Therefore, it is not appropriate to use RLDOCK to run VS.
2. The slope of rDock's AuROC-time curve in Figure 4b continues to increase. This raises the question of whether allocating more time to rDock (e.g., performing more exhaustive docking by conducting additional rDock runs) could potentially enable rDock's AuROC to match or even surpass the performance of RNAmigos++. The authors may want to comment on this question and provide insights into the potential impact of increased computational resources on rDock's performance relative to RNAmigos++.
3. In the Section "RNAmigos2 identifies riboswitch ligands in large-scale in vitro assay", the authors demonstrated the much better enrichment performance achieved by RNAmigos2, which combines predictions from different complementary models (i.e., aff and compat). It would be useful to show the enrichment performance of each individual model, i.e., aff and compat, to provide insights into their respective contributions to the overall performance.
4. While it is expected that combining different models would enhance overall performance, it is not clear how the model (compat) trained on experimental data enhances the success or complements the performance of the model (aff) trained on synthetic docking data.
5. The VS requires the structural information of the binding pocket. In many cases, the binding pocket structure is unknown or highly dynamic, it would be very helpful if the authors can provide some insights into this issue. For example, how sensitive is the prediction to the pocket structure information?

(Remarks on code availability)

Reviewer #2

(Remarks to the Author)

In this paper Carvajal-Patiño et al. have proposed a new augmented base machine learning approach to leverage the currently scarce available data on RNA small molecule and develop a virtual screening pipeline to overcome the limitation of traditional methods like molecular docking. There are several significant concerns about the approach and the results provided as outlined in the "questions and comments" sections. Overall, I do not recommend the publication of this paper in this form until all the questions and concerns are addressed.

Questions and comments:

1. in the introduction authors classify Autodock VINA as a "nucleic-acid-specific" suite.

The authors first reference to a paper which is a general description of the Vina package (there is no mentioning in this reference that this package is nucleic-acid specific) and then make an incorrect statement about the specificity of this package for nucleic acids which is inaccurate and misleading. They either need to provide supporting evidence for this claim (force field specifications of this docking package, scoring function adjustments for nucleic acids, internal parametrization, etc.) or remove this sentence.

2. Authors mention the high computational demands of the docking procedure (in the order of minutes) to justify the use of machine learning based approaches.

They should discuss that all the mentioned docking packages are CPU based suites. They need to discuss the GPU based docking packages that have tremendously decreased the time required for performing a single docking attempt (J. Chem. Theory Comput. 2021, 10.1021/acs.jctc.0c01006). Other docking packages have also been implemented in a way to take advantage of the computational speed up of GPUs. Authors need to discuss these also in the introduction to have a fair and scientific overview of the molecular docking approach.

3. Authors' argument about molecular docking.

There is a fundamental flaw with the argument authors are providing to justify the use of their approach. Their data augmentation process is totally dependent on the data provided by molecular docking without which the whole process does not have a noteworthy advantage compared to molecular docking.

They then in the Method section provide no details why they selected rDock (along with AutoDock Vina). How did they benchmarked the performance of the rDock, how successful is rDock in recapitulating the experimentally identified poses, what is the average accuracy of rDock (RMSD to heavy atoms), what is the accuracy of AutoDock Vina in this benchmarking, how accuracy changes with increasing the complexity of the ligands (increased degrees of rotations). DOCK 6 has also been benchmarked extensively for RNA small molecule complexes which authors do not mention (<http://www.rnajournal.org/cgi/doi/10.1261/rna.1563609>). Authors need to provide more details on the performance of rDock compared to, for example DOCK 6 and rationalize their choice of the rDock with solid evidence.

4. rDock outperforms the proposed model.

Authors fairly mention that rDock outperforms their model and then they try to argue that their model is advantages because of the running time. The point is accuracy plays a more important role than speed. Second, docking does not require a dataset for training and going through laborious processes of data preparation, splitting, optimization, training and benchmarking. Third, if the timing is the only advantage of the proposed approach, then authors need to compare their result with GPU based docking approaches.

5. authors claim, "We reach a performance of 0.990 AuROC to significantly outperform rDock (0.959)."

Based on what metric authors consider this difference "significant"? Authors need to be more careful with their wording in assessing the performance of their proposed approach. Second, as already mentioned the model is data augmented by using molecular docking. So overall despite all the implemented efforts there doesn't seem to be a noticeable merit of the RNAmigos2 to traditional docking.

6. "Hence, in a drug design setting and under a compute budget of one CPU day, docking enables the screening of about~1400 compounds while RNAmigos2 enables screening ~ 1.4M compounds".

Overall, the main argument authors provide in this paper is time-efficiency regarding the fact that docking has higher accuracy. I refer the authors to papers like "A practical guide to large-scale docking" (Nature Protocols volume 16, pages4799–4832 (2021)) which provide guidelines on how to perform docking on libraries with billions of compounds on moderately sized computer clusters with 500-1000 cores which considering the today's available resources is highly affordable. These highly optimized approaches have reduced the cost of docking to 1 s/molecule/core.

Minor revision:

Abstract (grammatical errors):

does not scale well with the size of the small molecule databases

typically using in data-driven strategies  typically using data-driven strategies

determined by the MinMaxPicker algorithm  determined

(Remarks on code availability)

Reviewer #3

(Remarks to the Author)

The manuscript by Carvajal-Patino, Mallet, et al. introduces a machine learning approach for in silico screening of small molecules targeting RNAs. The method performed better than state-of-the-art docking approaches both in terms of enrichment factor, i.e. ability to discriminate between active from inactive compounds, and computational cost. Given the exploding interest in small molecule targeting of RNAs from the academic community as well as the pharmaceutical companies, I believe that the approach proposed in this manuscript will be of interest to a wide readership while providing a tool that has the potential to significantly advance the field. However, there is a major issue and a few minor ones that I invite the authors to address before I can support the manuscript for publication in Nature Communications.

Major point

- In order to make the prediction of (pseudo) affinity of a ligand to an RNA target, RNAmigos2 requires structural information about the small molecule binding site, specifically an RNA structure along with a list of binding site nucleotides. All the results reported in the manuscript have been obtained given the exact knowledge of all the binding site residues. This naturally does not correspond to real life scenario. In many cases, the user provides a list of residues predicted by another in silico tool, which might not be 100% correct. I believe it is fundamental to test the proposed approach in real life scenarios. I therefore invite the authors to test RNAmigos2 using as input binding site residues predicted by one of the state-of-the-art approaches, such as SHAMAN (<https://www.nature.com/articles/s41467-024-49638-7>), which is currently the most accurate method, or BiteNet (similar accuracy of SHAMAN on holo-like conformations). Other methods, such as RBind or Sitemap, have been shown in [10.1038/s41467-024-49638-7](https://doi.org/10.1038/s41467-024-49638-7) to be less accurate than SHAMAN. My concern with RNAmigos2 is that it has been trained using experimental structures of RNA binding sites and therefore it might be significantly less accurate than traditional docking software when i) the input is only a partial list of (correct) binding site nucleotides or ii) when some residues provided as input are not part of the binding site, i.e. they are false positive predictions from an external software. While traditional docking software work decently well given a coarse-grained definition of the binding site region, i.e. a docking "box", it is currently unclear how much the performances of RNAmigos2 depend on the correct identification of all the binding site nucleotides.

Minor points

- Can the authors comment on the reason why in their training set they did not completely exclude experimental pockets populated by protein residues?

- It is unclear whether RNAmigos2 can also be used for blind docking, i.e. docking to an entire RNA structure without knowledge of the binding site residues. If not, the authors should mention this point in the "Discussion" part, and ideally test whether traditional docking software can instead obtain reasonable performances in blind docking. This is potentially an important point that would discourage using RNAmigos2 when there is no knowledge of the binding site nucleotides.

- It is unclear whether RNAmigos2 can provide structural information about the "docked" ligand. In some case, the bound conformation predicted by a docking software is of practical utility, for example to perform structure-based optimisation of a hit in absence of an experimental structure. If RNAmigos2 cannot provide this information, the authors should mention this limitation in the "Discussion" section of their manuscript.

(Remarks on code availability)

The code is well documented and the repository provides instructions about how to reproduce the results and figures reported in the manuscript

Version 1:

Reviewer comments:

Reviewer #1

(Remarks to the Author)

The authors have addressed this reviewer's original concerns.

(Remarks on code availability)

Reviewer #2

(Remarks to the Author)

The authors generally responded to our comments. The paper is ok, but not that great, but Nature Communications is probably an OK place for it.

(Remarks on code availability)

Reviewer #3

(Remarks to the Author)

In the revised version of their manuscript, the authors addressed satisfactorily all my previous concerns. I therefore support publication in the current form.

(Remarks on code availability)

The instructions to install and run the code, as well as reproduce the results presented in the manuscript, are provided in a clear fashion.

Response to the comments on : “RNAmigos2: Fast and accurate structure-based RNA virtual screening with semi-supervised graph learning and large-scale docking data”

Juan G. Carvajal-Patiño, Vincent Mallet, David Becerra, L. Fernando Niño V., Carlos Oliver, and Jérôme Waldispühl

We would like to thank the reviewers for their careful review of our manuscript and precise comments. It was highly encouraging to see that overall, reviewers were enthusiastic about the potential for our work to the broad community, with reviewer 1 indicating that the work “showcases the great potential of deep-learning techniques in modeling RNA-ligand interactions [...] and can be highly valuable to the scientific community”, and reviewer 3 remarked that, “this manuscript will be of interest to a wide readership while providing a tool that has the potential to significantly advance the field.” Concerns from the reviewers centered mainly on the following points:

- **Reviewer 1:** Deeper analysis to understand model performance regarding sub-models and sensitivity to binding site definitions.
- **Reviewer 2:** Enhancing docking benchmark, discussing relevance of ML methods in virtual screening.
- **Reviewer 3:** Similar to reviewer 1, the main concern was to understand model performance under imperfect binding site definitions.

In this revision, we carefully considered all comments from reviewers and proceeded to major changes based on these remarks which we believe have significantly strengthened the contribution. We have edited and rearranged the manuscript with all the reviewers comments in mind and hope it is to the reviewers satisfaction. Changes in the manuscript are highlighted in blue. Before diving into the details, we would like to summarize the three main changes in this revised manuscript:

Update 1: Following reviewer 2’s suggestion, we included DOCK6 in our pose selection and virtual screening benchmark.

Update 2: Thanks to the reviewers comments, we identified a subtle flaw in our benchmark on the ROBIN data set. Inference was performed independently on the actives and inactives, which resulted in independent normalizations and ultimately inflated performances. We fixed this error, re-train our model, and now report more accurate results, which are thankfully more consistent with the rest of the results of our paper.

Update 3: Based on requests from reviewers 1 and 3, we included a pocket sensitivity assessment. We determined that the models included in our initial submission were eventually too robust, and thus not specific enough. This can happen if a large part of the performance is

attributable to QSAR (ligand-driven screening), and was previously documented on protein-ligand affinity prediction⁶. Hence, we updated our models and validation for enhanced specificity in the following way:

- **Model:** We added a margin-based loss to the training of the Compat model. This was designed to enhance the pocket-specificity of our predictions by maximizing the margin between true interactions and non-specific ones’.
- **Evaluation:** We additionally report the performance of our models when performing inference with shuffled pairs to determine target specificity.
- **Evaluation:** We added a pocket sensitivity assessment, with different perturbation strategies, taking into account partial detection at the right localization (missing or extra nodes), or at a shifted localization. We are able to retain most of our performance, which indicates a potential integration with existing pocket mining tools.
- **Analysis:** We added more analysis on the complementarity of our tool, showing that training on margins allows for the discovery of more diverse compounds. To this end, visualization of chemical spaces is also simplified (Fig 6d) to allow for better understanding of model behaviour and complementarity.

Importantly, those updates are marginally impacting the initially reported performance of RNAmigos2, but they are substantially reinforcing the robustness and specificity of our results.

In summary, even though the magnitude of the performance on the ROBIN data set is tempered, it remains significant and, in retrospect, more consistent with the results reported in the rest of the manuscript. Overall, we believe the revised analysis reinforces our trust in the significance and robustness of the results. Furthermore, this revision allowed us introduce models with enhanced pocket specificity, and to develop a rigorous and comprehensive validation scheme of the robustness of RNAmigos to imperfect pocket detection.

We would like to reiterate that the core contributions of RNAmigos2 remain valid:

- A stand-alone performance similar to docking on strict test sets, yielding a massive speedup over traditional docking (5ms instead of 1s-1min per compound)
- A synergistic integration with docking, acting as a correction term and eventually cutting its error by four in a quarter of the time
- A milestone in the development of machine-learning based virtual screening methods on RNA, introducing with a highly relevant data split and validation pipeline that includes large scale experimental screening data.

We thus believe our method is a milestone for further machine-learning based methods developments for RNA drug discovery.

We also would like to clarify that our contribution does not aim to replace molecular docking but rather to help accelerating current pipelines. Our method can be seen as an excellent pre-screening tool for virtual screening applications; but for instance, still entirely rely on molecular docking for pose generation. Moreover, it can synergistically integrate with existing tools, resulting in mixed models, as is detailed in section *RNAmigos predictions consistently boost virtual screening*

efficiency.

Again, we sincerely thank the reviewers for their help in considerably improving the robustness and quality of our contribution.

Comments by Reviewer 1

In this manuscript, the authors proposed a structure-based machine learning pipeline, termed RNAmigos2, for fast and accurate large-scale virtual screening of compounds against RNA targets. The model employs an encoder-decoder framework and leverages graph representations for both RNA and ligands. The RNA and ligand encoders are pre-trained using unsupervised graph learning on extensive RNA and ligand structural datasets. Subsequently, the downstream decoders are trained by utilizing both experimental and synthetic data collected from PDB and docking software to identify native binding ligand from decoys and reproduce the docking scores, respectively. A key feature of the proposed approach is the massive speedup achieved over traditional docking-based approaches while still obtaining higher enrichment performance in RNA-targeted virtual screening. Notably, the authors provide detailed steps on GitHub for reproducing the manuscript's results. This work showcases the great potential of deep-learning techniques in modeling RNA-ligand interactions, particularly in structure-based virtual screening workflows, and can be highly valuable to the scientific community.

We thank the reviewer for this summary of the context and contribution of this work.

In Section \RNAmigos2 is competitive with docking software and generalizes to new targets". The RLDOCK model provides a nice scoring function for R-L docking, but the model is not designed for VS because RLDOCK contains a ligand intramolecular LJ energy term that cannot be directly compared between different compounds. Therefore, it is not appropriate to use RLDOCK to run VS.

We thank the reviewer for highlighting this inaccuracy, which could explain the poor performance of RLDOCK under this virtual screening metric. Thus, we recompute RLDOCK performance while removing the ligand and receptor self-energy terms and we do observe a modest improvement in AuROC from 66 to 67. We updated our results.

The slope of rDock’s AuROC–time curve in Figure 4b continues to increase. This raises the question of whether allocating more time to rDock (e.g., performing more exhaustive docking by conducting additional rDock runs) could potentially enable rDock’s AuROC to match or even surpass the performance of RNAmigos++. The authors may want to comment on this question and provide insights into the potential impact of increased computational resources on rDock’s performance relative to RNAmigos++.

In Figure 4b, we report the maximum time needed to score all compounds with docking software. Therefore, we cannot improve the performance by allowing more time to docking. The confusion probably stems from the y-logarithmic scale we used. Without this y-scale, we would see a smooth increase between a random behavior (AuROC of 0.5 before screening) and final performance of rDock after screening all the compounds (95.9). The log-scale on the y-axis is useful to see the difference between RNAmigos and RNAmigos++. However, we acknowledge that it could (incorrectly) suggest that the slope of the docking performance increases at the end of the screening. We added a clarification in the text commenting those results and the caption of the plot.

In the Section "RNAmigos2 identifies riboswitch ligands in large-scale in vitro assay", the authors demonstrated the much better enrichment performance achieved by RNAmigos2, which combines predictions from different complementary models (i.e., aff and compat). It would be useful to show the enrichment performance of each individual model, i.e., aff and compat, to provide insights into their respective contributions to the overall performance.

We thank reviewer 1 for this remark, as it led to gaining insights about our models. We included those results in Supplementary Table 3 and a visual analysis of the performance of those models in Supplementary Figure 9. As can be seen, our updated models are complementary in the sense that they do not retrieve the same actives: Aff model focuses on highly populated clusters, resulting in high AuROCs but at the cost of a reduced specificity. On the other hand, our updated Compat model learns to focus on pocket-specific compounds. We have included a discussion of this result in the text and point the reviewer to the last section of the Results for a closer analysis of the complementarity of models.

While it is expected that combining different models would enhance overall performance, it is not clear how the model (compat) trained on experimental data enhances the success or complements the performance of the model (aff) trained on synthetic docking data.

Ensembles of learnt models have been repeatedly shown to enhance performance². This is illustrated in our case since a single Compat model gives an AuROC of 84.4, while an ensemble of two Compat models gives 89.7 (Table 1). However, it was not obvious that the ensemble of two different models (97.2) would outperform the ensemble of a single modality.

As an explanation for this synergy, we report a complementarity of the observed error modes in

Figure 2.D, where we observe that target where the Aff or rDock predictions fail, are in several cases correctly predicted by Compat and vice versa. However, we acknowledged that we did not offer an interpretation for this complementarity, and thus on page 5, we added a sentence in the text: "This complementarity might originate from force field priors being too strong to accurately fit the data. Our averaging is reminiscent of Bayesian fitting, with a likelihood term and a physical prior term." With regards to the analysis of complementarity of our results, we kindly refer reviewer 1 to our answer to the previous question.

The VS requires the structural information of the binding pocket. In many cases, the binding pocket structure is unknown or highly dynamic; it would be very helpful if the authors can provide some insights into this issue. For example, how sensitive is the prediction to the pocket structure information?

Reviewer 3 had a similar inquiry. We added an experiment to assess our tool's robustness to pockets being only approximately right. We randomly perturbed our binding site's definition to include between 0.5 and 1.35 times the number of residues of the ground truth binding site. This value was chosen based on the state-of-the-art binding site methods' recall. We randomly subsample a corresponding amount of nodes in the neighborhood of ground truth binding site, following two strategies (imperfect pocket selection, and randomized pocket location), resulting in several sets of perturbed inputs to our methods with varying degree of perturbation.

We show that under these two scenarios of binding site detections, the validation obtained lower albeit similar performance, indicating that our tool can efficiently be used on the output of binding site prediction tools.

Comments by Reviewer 2

In this paper Carvajal-Patiño et al. have proposed a new augmented base machine learning approach to leverage the currently scarce available data on RNA small molecule and develop a virtual screening pipeline to overcome the limitation of traditional methods like molecular docking. There are several significant concerns about the approach and the results provided as outlined in the "questions and comments" sections. Overall, I do not recommend the publication of this paper in this form until all the questions and concerns are addressed.

We thank reviewer 2 for his or her review. We tried to address all his or her questions and concerns and hope it will meet his or her expectations. We identified three major themes in the reviewer's questions:

- docking procedure validation
- speed considerations
- performance considerations.

In order to provide a clear and comprehensive response, we grouped our answers accordingly, and provide them below:

1. In the introduction authors classify Autodock VINA as a "\nucleic-acid-specific" suite. The authors first reference to a paper which is a general description of the Vina package (there is no mentioning in this reference that this package is nucleic-acid specific) and then make an incorrect statement about the specificity of this package for nucleic acids which is inaccurate and misleading. They either need to provide supporting evidence for this claim (force field specifications of this docking package, scoring function adjustments for nucleic acids, internal parametrization, etc.) or remove this sentence.

3. [...] They then in the Method section provide no details why they selected rDock (along with AutoDock Vina). How did they benchmarked the performance of the rDock, how successful is rDock in recapitulating the experimentally identified poses, what is the average accuracy of rDock (RMSD to heavy atoms), what is the accuracy of AutoDock Vina in this benchmarking, how accuracy changes with increasing the complexity of the ligands (increased degrees of rotations). DOCK 6 has also been benchmarked extensively for RNA small molecule complexes which authors do not mention (<http://www.rnajournal.org/cgi/doi/10.1261/rna.1563609>). Authors need to provide more details on the performance of rDock compared to, for example DOCK 6 and rationalize their choice of the rDock with solid evidence.

We agree that AutodockVina is not nucleic-acid-specific and therefore we remove this sentence from the text. However we note that the Luo et al.³ review points out AutoDock has been used without modifications on nucleic acids.

In the initial submission, we conducted an internal benchmark to select a docking tool and included the results in supplementary Section D. We noted our initial paper was missing a pointer to this supplementary material section in the main text. In the revised manuscript, we added a reference to this section in the Methods section (Docking Dataset Construction). Furthermore, we added a note in the discussion to emphasize that other choices of docking tools are possible, and that our model architecture is fully compatible with them. We also suggested there to explore of the impact on other programs in future works.

Finally, to measure DOCK6's performance and compare it with rDock, we included DOCK 6 in the software evaluation presented in supplementary Section D. Additionally, we performed docking experiments with DOCK 6 on the test data set, obtaining better results with rDock (0.82 vs 0.96 AuROC).

2. Authors mention the high computational demands of the docking procedure (in the order of minutes) to justify the use of machine learning based approaches. They should discuss that all the mentioned docking packages are CPU based suites. They need to discuss the GPU based docking packages that have tremendously decreased the time required for performing a single docking attempt (J. Chem. Theory Comput. 2021, 10.1021/acs.jctc.0c01006). Other docking packages have also been implemented in a way to take advantage of the computational speed up of GPUs. Authors need to discuss these also in the introduction to have a fair and scientific overview of the molecular docking approach.

6. "Hence, in a drug design setting and under a compute budget of one CPU day, docking enables the screening of about 1400 compounds while RAmigos2 enables screening 1.4M compounds". Overall, the main argument authors provide in this paper is time-efficiency regarding the fact that docking has higher accuracy. I refer the authors to papers like "A practical guide to large-scale docking" (Nature Protocols volume 16, pages4799-4832 (2021)) which provide guidelines on how to perform docking on libraries with billions of compounds on moderately sized computer clusters with 500-1000 cores which considering the todays available resources is highly affordable. These highly optimized approaches have reduced the cost of docking to 1 s/molecule/core.

We thank the reviewer for the pointers on fast alternatives to classical docking, namely GPU-based acceleration and fast, less accurate docking. We added references to these alternatives, as well as a short discussion in the introduction.

We use this opportunity to briefly comment on these two avenues as well as clarifying (if needed) of the motivation and justification of our contribution. Modern GPU-based docking reports runtimes of about 0.1 GPU s/compound to compute a docking score, and are able to match classical docking accuracy. However, setting up a highly parallel experiment on a GPU cluster is not as straightforward as using a CPU cluster. Moreover, the price of a CPU is also in the order of 100 times cheaper than a GPU. Finally, this time benchmark does not take into account ligand preparation. This could explain why in the Nature Protocol paper enclosed (published 3 years after GPU docking was published), there is no mention of GPU-based approaches.

The other avenue is to use a less precise, faster docking procedure. From Bender et al.¹ introduction :

To be feasible for a billion-molecule library on moderately sized computer clusters (e.g., 500–1,000 cores), this calculation must consume not much more than 1 s/molecule/core (1 ms/configuration). This need for speed means that the calculation cannot afford the level of detail and number of interaction terms that would be necessary to achieve chemical accuracy. For instance, docking typically undersamples conformational states, ignores important terms (e.g., ligand strain) and approximates terms that it does include (e.g., fixed potentials) Owing to these approximations and neglected terms, docking energies have known errors, and the method cannot even reliably rank order molecules from a large library screen. What it can hope to do, however, is separate a tiny fraction of plausible ligands from the much larger number of library molecules that are unlikely to bind a target.

In summary, a lower accuracy, faster docking can serve as a rough scanning tool running in

approximately 1 core s/compound. Moreover, this reported time represents one of a long series of steps (about 100), including some other relatively time-consuming ones. They refer to actual virtual screening campaigns, where the reported time for screening varied from 1s to about two minutes per compound.

Our approaches are orders of magnitude faster. We initially did not extensively optimize them since computation time is negligible in comparison with docking. However, out of completeness, in this revision we report an optimized computation time of 5.8 core ms/compound on raw smiles, and 1.6 core ms/compound on precomputed graphs (which is a fairer comparison, since the Nature Protocol paper uses precomputed conformers).

3. Authors' argument about molecular docking. There is a fundamental flaw with the argument authors are providing to justify the use of their approach. Their data augmentation process is totally dependent on the data provided by molecular docking without which the whole process does not have a noteworthy advantage compared to molecular docking. [...]

4. rDock outperforms the proposed model. Authors fairly mention that rDock outperforms their model and then they try to argue that their model is advantages because of the running time. The point is accuracy plays a more important role than speed. Second, docking does not require a dataset for training and going through laborious processes of data preparation, splitting, optimization, training and benchmarking. Third, if the timing is the only advantage of the proposed approach, then authors need to compare their result with GPU based docking approaches.

5. authors claim, "We reach a performance of 0.990 AuROC to significantly outperform rDock (0.959)." Based on what metric authors consider this difference "significant"? Authors need to be more careful with their wording in assessing the performance of their proposed approach. Second, as already mentioned the model is data augmented by using molecular docking. So overall despite all the implemented efforts there doesn't seem to be a noticeable merit of the RNAmigos2 to traditional docking.

We mention that rDock outperforms our single-modality models, but we then show that the ensemble average of our models results in a better average performance than rDock. We apologize for using the word significant without adding a statistical analysis backing this result. A Whitney-Houston U-test enabled us to compare the performance of MIXED and the one of rDock, and resulted in a p-value of 0.0068. We included this information in our revised manuscript (revised manuscript, top of page 6) .

We agree that docking does not require any training procedure (even though it requires careful preparation steps), but we took care of training this model for the community, and users can now access the results of our training effortlessly. We underline that users do not have to do any of the steps you mention (preparation, splitting, training...). Additionally, they do not have to go through the steps of ligand and pocket preparation that are necessary to docking to use our tool, and benefit from extremely fast computations. We also recall that unlike many docking software, RNAmigos2 only requires a low-resolution representation of the RNA structure (i.e., non-canonical base pair interaction network), which opens up the tool for use with coarse-grained structure prediction methods⁵.

Finally, we fully acknowledge that the model is data augmented by using molecular docking, that is in fact a cornerstone of our contribution. We do not claim (and do not want to) that docking is being replaced by our methods. This is precisely why we also report enhanced results by actually running docking on the top ranked compounds of RNAmigos (the RNAmigos++ scenario in Fig 4 and 5). We do see our tool as complementary to docking. We deeply support the development of docking tools as a contemporary line of work, and believe these enhanced tools will generate higher quality data for our tool to be trained on, as now included in the discussion.

Comments by Reviewer 3

The manuscript by Carvajal-Patino, Mallet, et al. introduces a machine learning approach for in silico screening of small molecules targeting RNAs. The method performed better than state-of-the-art docking approaches both in terms of enrichment factor, i.e. ability to discriminate between active from inactive compounds, and computational cost. Given the exploding interest in small molecule targeting of RNAs from the academic community as well as the pharmaceutical companies, I believe that the approach proposed in this manuscript will be of interest to a wide readership while providing a tool that has the potential to significantly advance the field. However, there is a major issue and a few minor ones that I invite the authors to address before I can support the manuscript for publication in Nature Communications.

We thank the reviewer for his or her thoughtful comments and assessment on the potential for a broad impact of this contribution.

In order to make the prediction of (pseudo) affinity of a ligand to an RNA target, RNAmigos2 requires structural information about the small molecule binding site, specifically an RNA structure along with a list of binding site nucleotides. All the results reported in the manuscript have been obtained given the exact knowledge of all the binding site residues. This naturally does not correspond to real life scenarios. In many cases, the user provides a list of residues predicted by another in silico tool, which might not be 100% correct. I believe it is fundamental to test the proposed approach in real life scenarios. I therefore invite the authors to test RNAmigos2 using as input binding site residues predicted by one of the state-of-the-art approaches, such as SHAMAN (<https://www.nature.com/articles/s41467-024-49638-7>), which is currently the most accurate method, or BiteNet (similar accuracy of SHAMAN on holo-like conformations). Other methods, such as RBind or Sitemap, have been shown in [10.1038/s41467-024-49638-7](https://doi.org/10.1038/s41467-024-49638-7) to be less accurate than SHAMAN. My concern with RNAmigos2 is that it has been trained using experimental structures of RNA binding sites and therefore it might be significantly less accurate than traditional docking software when i) the input is only a partial list of (correct) binding site nucleotides or ii) when some residues provided as input are not part of the binding site, i.e. they are false positive predictions from an external software. While traditional docking software work decently well given a coarse-grained definition of the binding site region, i.e. a docking ‘‘box’’, it is currently unclear how much the performances of RNAmigos2 depend on the correct identification of all the binding site nucleotides.

Reviewer 1 had a similar inquiry. We added a new experiment to assess the robustness of our prediction with respect to the precision of the pocket location and definition. We perturbed our binding site’s definition to include between 0.5 and 1.35 times the number of residues of the original binding site. This value was chosen based on the state-of-the-art binding site methods’ recall. We therefore randomly sample a percentage of nodes in the neighborhood of the ground truth binding site. Then, we follow two strategies(noised and shifted pockets) that result in a collection of approximate binding sites with increasing levels of perturbations. We show in Fig 3 c and d, that under two plausible scenarios of binding site detection uncertainty, the validation obtained lower albeit similar performance, indicating that our tool can reliably be used on the output of binding site prediction tools.

Minor point:

Can the authors comment on the reason why in their training set they did not completely exclude experimental pockets populated by protein residues?

Many RNAs are crystallized in the vicinity of protein chains. Taking only RNAs in isolation would result in significantly reduced dataset sizes. To balance this tradeoff, we use a filter previously introduced⁴, in which we define the binding site as all nucleotides and residues within a sphere of 10 Å radius. Within that volume, we compute the percentage of RNA and protein residues, removing any putative binding sites with more than 40% protein residues. We have made this clarification in

the revised manuscript under the Methods subsection, “Constructing and representing a Dataset of Binding Sites”.

It is unclear whether RNAmigos2 can also be used for blind docking, i.e. docking to an entire RNA structure without knowledge of the binding site residues. If not, the authors should mention this point in the "Discussion" part, and ideally test whether traditional docking software can instead obtain reasonable performances in blind docking. This is potentially an important point that would discourage using RNAmigos2 when there is no knowledge of the binding site nucleotides.

Our previous experiments (Fig 3 c, d) suggest that our method can be used in conjunction with a binding site prediction algorithms, and cope with inaccurate predictions. Blind docking still under-performs docking and reports longer runtime (authors do not report hardware configuration), for instance see `cbdock2`? . We believe the most favorable results are still obtained by using two tools. It is an interesting direction to integrate both steps in a blind-docking framework, which was already successfully applied in proteins, but it is unclear whether there is enough biological signal in the available RNA data. We thank the reviewer for this comment and now mention this limitation and future direction in the discussion.

It is unclear whether RNAmigos2 can provide structural information about the "docked" ligand. In some cases, the bound conformation predicted by a docking software is of practical utility, for example to perform structure-based optimisation of a hit in absence of an experimental structure. If RNAmigos2 cannot provide this information, the authors should mention this limitation in the "Discussion" section of their manuscript.

This is indeed a limitation of the RNAmigos tool. However, using the RNAmigos++ pipeline, top hits are assessed using docking, which thus generates docking pose. Overall, we see our tool as a way to complement and accelerate docking, thus retaining this aspect of its advantages.

The code is well documented and the repository provides instructions about how to reproduce the results and figures reported in the manuscript

References

- [1] Brian J Bender, Stefan Gahbauer, Andreas Lutten, Jiankun Lyu, Chase M Webb, Reed M Stein, Elissa A Fink, Trent E Balius, Jens Carlsson, John J Irwin, et al. A practical guide to large-scale docking. *Nature protocols*, 16(10):4799–4832, 2021.

- [2] Mudasir A Ganaie, Minghui Hu, Ashwani Kumar Malik, Muhammad Tanveer, and Ponnuthurai N Suganthan. Ensemble deep learning: A review. *Engineering Applications of Artificial Intelligence*, 115:105151, 2022.
- [3] Jiaying Luo, Wanlei Wei, Jérôme Waldispühl, and Nicolas Moitessier. Challenges and current status of computational methods for docking small molecules to nucleic acids. *European journal of medicinal chemistry*, 168:414–425, 2019.
- [4] Carlos Oliver, Vincent Mallet, Roman Sarrazin Gendron, Vladimir Reinharz, William L Hamilton, Nicolas Moitessier, and Jérôme Waldispühl. Augmented base pairing networks encode rna-small molecule binding preferences. *Nucleic acids research*, 48(14):7690–7699, 2020.
- [5] Xiujuan Ou, Yi Zhang, Yiduo Xiong, and Yi Xiao. Advances in rna 3d structure prediction. *Journal of Chemical Information and Modeling*, 62(23):5862–5874, 2022.
- [6] Mikhail Volkov, Joseph-André Turk, Nicolas Drizard, Nicolas Martin, Brice Hoffmann, Yann Gaston-Mathé, and Didier Rognan. On the frustration to predict binding affinities from protein–ligand structures with deep neural networks. *Journal of medicinal chemistry*, 65(11):7946–7958, 2022.